# Towards imaging of atmospheric trace gases using Fabry Pérot Interferometer Correlation Spectroscopy in the UV and visible spectral range

Jonas Kuhn[1,2], Ulrich Platt[1,2], Nicole Bobrowski[1,2], and Thomas Wagner[2]

[1]Institute of Environmental Physics, University of Heidelberg, Germany
[2]Max Planck Institute for Chemistry, Mainz, Germany

**Correspondence:** J. Kuhn (jkuhn@iup.uni-heidelberg.de)

**Abstract.** Many processes in the lower atmosphere including transport, turbulent mixing and chemical conversions happen on time scales of the order of seconds (e.g. at point sources). Remote sensing of atmospheric trace gases in the UV and visible spectral range (UV/Vis) commonly uses dispersive spectroscopy (e.g. Differential Optical Absorption Spectroscopy, DOAS). The recorded spectra allow for the direct identification, separation and quantification of narrow band absorption of trace gases. However, these techniques are typically limited to a single viewing direction and limited by the light throughput of the spectrometer setup. While two dimensional imaging is possible by spatial scanning, the temporal resolution remains poor (often several minutes per image). Therefore, processes on time scales of seconds cannot be directly resolved by state of the art dispersive methods.

We investigate the application of Fabry- Pérot Interferometers (FPIs) for the optical remote sensing of atmospheric trace gases in the UV/Vis. By choosing a FPI transmission spectrum, which is optimised to correlate with narrow band (ideally periodic) absorption structures of the target trace gas, column densities of the trace gas can be determined with a sensitivity and selectivity comparable to dispersive spectroscopy, using only a small number of spectral channels (FPI tuning settings). Different from dispersive optical elements, the FPI can be implemented in full frame imaging setups (cameras), which can reach high spatio-temporal resolution. In principle, FPI Correlation Spectroscopy can be applied for any trace gas with distinct absorption structures in the UV/Vis.

We present calculations for the application of FPI Correlation Spectroscopy to $SO_2$, $BrO$ and $NO_2$ for exemplary measurement scenarios. Besides high sensitivity and selectivity we find that the spatio temporal resolution of FPI Correlation Spectroscopy can be more than two orders of magnitude higher than state of the art DOAS measurements.

As proof of concept we built a one-pixel prototype implementing the technique for $SO_2$ in the UV. Good agreement with our calculations and conventional measurement techniques are demonstrated and no cross sensitivities to other trace gases are observed.

# 1 Introduction

Within the last decades, progress in optical remote sensing of atmospheric trace gases led to a better understanding of many important processes including air pollution, ozone and halogen chemistry, and the evolution of volcanic plumes. Narrow band structures in the trace gas molecule's absorption spectrum are used to identify and quantify the amount of a trace gas integrated along a line of sight, i.e. its column density (CD, typically in units of $\mathrm{molec\,cm^{-2}}$) and to separate its absorption signal from interfering gas absorptions and scattering processes.

Differential Optical Absorption Spectroscopy (DOAS, see Platt and Stutz, 2008, for details) has become a well established technique for atmospheric trace gas remote sensing in the UV/Vis with high sensitivity (detection limits within the ppb to ppt range for atmospheric light paths of a few kilometers). A spectrometer and a telescope with narrow field of view (FOV) are used to record spectra $I(\lambda)$ of scattered sunlight, which are compared to a reference spectrum $I_0(\lambda)$. Beer Lambert's law describes the corresponding spectral optical density $\tau(\lambda)$ with $\sigma_i(\lambda)$ and $c_i(l)$ being the absorption cross section and the concentration of trace gas species $i$ along a line of sight $L$, respectively:

$$\tau(\lambda) = -\log\frac{I(\lambda)}{I_0(\lambda)} = \int_0^L \sum_i \sigma_i(\lambda)c_i(l)\mathrm{d}l + \text{scattering at molecules and aerosols} \tag{1}$$

Atmospheric UV/Vis optical densities are dominated by trace gas absorption and scattering processes at air molecules or aerosols. The known absorption cross sections of the trace gases together with a polynomial, which accounts for the broad band absorption and scattering effects are fitted to the measured spectral optical density. The fit coefficients represent the CDs $S_i = \int_0^L c_i(l)\mathrm{d}l$, i.e. the integrated trace gas concentration along the light path difference of $I(\lambda)$ and the reference $I_0(\lambda)$. In principle, spatial distributions of trace gases (images or height profiles) can be recorded by scanning of viewing angles with a narrow FOV telescope (e.g. Multi-Axis DOAS, Hönninger et al., 2004). More complicated optical setups allow to record spectra of an entire image column at once by using a two dimensional detector array (e.g. Imaging DOAS, Lohberger et al., 2004). Images can then be recorded by column (push broom) scanning. The acquisition times of these techniques used to be are rather large (often several minutes per image or profile for typical trace gas CDs) limiting their application to processes that are spatially homogeneous on that time scale or processes with very high trace gas CDs. Recently, Manago et al. (2018) reported $NO_2$ measurements with an hyperspectral camera based on the Imaging DOAS technique with considerably higher spatial resolution ($\sim 0.08\,\mathrm{Hz}$).

Determination of two dimensional atmospheric trace gas distributions with high time resolution at time scales of the order of seconds, i.e. fast imaging of atmospheric trace gases is possible with techniques recording all spatial pixels of an image at once for a low number of spectral channels (see e.g. Platt et al., 2015). This allows studying phenomena which are not accessible to conventional scanning methods. With fast trace gas imaging techniques, sources and sinks of trace gases can be identified and quantified on much smaller spatial and temporal scales than with conventional remote sensing techniques. This allows for instance to gain insight into small scale mixing processes and to distinguish chemical conversions from transport.

Most of the presently used atmospheric trace gas imaging schemes either use a set of two band pass filters (e.g. $SO_2$ Camera, $\sim 1\,\mathrm{Hz}$ for volcanic emissions, see e.g. Mori and Burton, 2006; Bluth et al., 2007; Kern et al., 2010; Platt et al., 2018) or a

tuneable band pass filter as wavelength selective element (e.g. $NO_2$ Camera, $\sim 3\,\mathrm{min}$ per image for stack emissions of power plants Dekemper et al., 2016). These techniques either involve intricate optical set-ups with low light throughput or yield a rather coarse spectral resolution which might result in strong cross interferences (see e.g. Lübcke et al., 2013; Kuhn et al., 2014).

Here we study the application of Fabry Pérot Interferometers (FPIs) as wavelength selective elements for trace gas imaging in the UV/Vis. The periodic FPI transmission spectrum is matched to the spectral absorption structures of trace gases that often show a similar periodicity. Thanks to the high correlation of the transmission spectrum of the wavelength selective element and the trace gas absorption spectrum a high sensitivity can be reached and cross interferences with other absorbers are minimized, even if only a small number of spectral channels (FPI tuning settings) is used (see Kuhn et al., 2014, and

the discussion in Sect. 3.2 below). Air spaced FPI etalons are very robust devices and allow for simple optical designs that can easily be implemented in imaging sensors. We present a model study on the sensitivity and selectivity of FPI Correlation Spectroscopy applied to $SO_2$, BrO and $NO_2$ (Sect. 3). For exemplary measurement scenarios, we infer the possible spatio-temporal resolution of these measurements for a specific instrument implementation. We find that, for the three gases, imaging with spatio-temporal resolutions about two orders of magnitude higher than state of the art methods should be possible. In

addition, we present a proof of concept study of the technique for volcanic sulphur dioxide ($SO_2$), validating the expected high accuracy and sensitivity of the technique with a one pixel prototype instrument. ==Further, we show that $SO_2$ CDs can be accurately retrieved from the recorded data without calibration (Sect. 4).==

## 2   Fabry Pérot Interferometer Correlation Spectroscopy

### 2.1   Fabry Pérot Interferometer

The FPI is a fundamentally very simple optical device, known for more than a century (e.g. Perot and Fabry, 1899). In principle, it consists of two plane parallel surfaces each with reflectance $R$, separated by a distance $d$ (see Fig. 1). The medium between the plates has the index of refraction $n$. Incident light (angle of incidence $\alpha$) is split up in partial beams with different optical path lengths between the two surfaces. Due to interference of the transmitted partial beams, the spectral transmission of the FPI is characterised by periodic transmission peaks, referring to constructive interference. For high enough orders of

interference, the free spectral range (FSR) $\Delta_\lambda$ between two transmission peaks in units of wavelength $\lambda$ is approximately given by:

$$\Delta_\lambda(\lambda) \approx \frac{\lambda^2}{2nd\cos\alpha} \qquad (2)$$

The finesse $F$ of a FPI represents the ratio of FSR to the full width half maximum (FWHM) $\delta_\lambda$ of a transmission peak:

$$F = \frac{\Delta_\lambda}{\delta_\lambda} \qquad (3)$$

The finesse is a measure for the number of effectively interfering partial beams and therefore increasing with the surface reflectance. However, it is also depending on the alignment and quality of the surfaces.

The spectral transmission as a function of $\lambda$ and the FPI's instrument parameters is given by:

$$T_{FPI}(\lambda) = \left[ 1 + \frac{4R}{(1-R)^2} \sin^2 \left( \frac{2\pi dn \cos \alpha}{\lambda} \right) \right]^{-1} \tag{4}$$

Despite its simple design, the challenge in manufacturing FPI devices lies in creating set-ups keeping $d$ stable to a fraction of a wavelength across the effective aperture.

## 2.2 Detection principle

The concept of using FPI correlation to detect atmospheric trace gases is described in Kuhn et al. (2014). The correlation of periodic absorption structures of atmospheric trace gases and the FPI transmission is exploited. An apparent absorbance $\tilde{\tau}_i$ of a trace gas $i$ is calculated from the optical densities of an on-band $\tau_A$ and off-band $\tau_B$ channel:

$$\tilde{\tau}_i = \tau_A - \tau_B = \log \frac{I_{A,0}}{I_A} - \log \frac{I_{B,0}}{I_B} = (\overline{\sigma}_{A,i} - \overline{\sigma}_{B,i}) S_i = \Delta\overline{\sigma}_i S_i \tag{5}$$

$S_i$ denotes the CD of a trace gas $i$. For the on-band channel, the spectral pattern of the FPI transmittance is chosen to correlate with the absorption band structure of the target trace gas, while for the off-band channel the FPI is tuned to show minimum correlation with the target trace gas absorption (see Fig. 1). The apparent absorbance is - for low trace gas optical densities - proportional to the CD of the trace gas. The proportionality is $\Delta\overline{\sigma}$, representing the difference of the effective absorption cross section seen by channel A and channel B. The optical densities $\tau_k$ are calculated from measured radiances $I_k$ transmitted by the FPI in a setting $k = A, B$:

$$I_k = \int_{\Delta\lambda} I(\lambda) \cdot T_{FPI,k}(\lambda) \, d\lambda \tag{6}$$

$I_{k,0}$ denotes the reference radiance without the target trace gas in the light path. In practice, a wavelength range $\Delta\lambda$ of high correlation of spectral trace gas absorption and FPI transmission is preselected with a band pass filter (BPF). Within this spectral range the FPI physical parameters are optimised. Figure 1c shows the optical density of $BrO$ seen through an FPI with varying surface displacement $d$. The maximum difference between maximum and minimum optical density determine the FPI settings A and B. In addition, the Finesse is chosen to maximise the signal to noise ratio.

Here we apply FPI Correlation Spectroscopy for passive imaging of a trace gas in the atmosphere. This means that the light source is scattered sky radiation that is measured within an imaging FOV (e.g. 20° aperture angle). We assume in the following that a reference $I_0$ (i.e. a part without trace gas) is always present within the image, so that $S$ denotes the differential trace gas CD compared to that reference.

The proportionality $\Delta\overline{\sigma}$ of apparent absorbance $\tilde{\tau}$ and trace gas CD $S$ (see Eq. 5) can be calculated from literature absorption cross sections of the target trace gas, a background spectrum $I_0(\lambda)$ and a modeled instrument transfer function (see Sect. 3.1 and 4.2). Alternatively, $\Delta\overline{\sigma}$ can be determined through calibration (see e.g. Lübcke et al., 2013; Sihler et al., 2017).

## 3  Model study for $SO_2$, $NO_2$ and BrO

FPI Correlation Spectroscopy can be applied to every trace gas species that yields sufficiently strong spectral absorption structures in the regarded wavelength range. In this section we present exemplary model studies for imaging of $SO_2$, $NO_2$ and BrO. For each target trace gas we regard a typical exemplary measurement scenario (Tab. 1). For the target species BrO and $SO_2$ we use typical measurement scenarios of volcanic emissions in the UV spectral range. For $SO_2$ we assume CDs that are typically measured in volcanic plumes of a comparably weak volcanic emitter or an already highly diluted plume (required detection limit of 1e17 $\mathrm{molec\,cm^{-2}}$, see Tab. 1). We additionally chose a high and probably disturbing $NO_2$ CD (1e17 $\mathrm{molec\,cm^{-2}}$) in oder to make the scenario also applicable to $SO_2$ measurements at e.g. ship or industrial stack emissions. Existing filter based $SO_2$ cameras are subject to strong cross interferences in this CD range (see e.g. Lübcke et al., 2013; Kuhn et al., 2014). In the scenario for BrO we assume a relatively strong but not uncommon volcanic emitter, with BrO mixing ratios of tens to hundreds ppt within the plume (required detection limit of 1e14 $\mathrm{molec\,cm^{-2}}$, see Tab. 1) and high $SO_2$ CDs (3e18 $\mathrm{molec\,cm^{-2}}$). Gradients in the BrO distributions can give insight into in-plume halogen chemistry (see e.g. Bobrowski et al., 2007; von Glasow, 2010; Roberts et al., 2014). The $NO_2$ scenario (blue spectral range) is applicable to measurements of stack emissions at e.g. a coal power plant (see e.g. Dekemper et al., 2016) but also to local gradients induced by traffic (required detection limit of 1e16 $\mathrm{molec\,cm^{-2}}$, see Tab. 1).

We calculate the sensitivities and study the cross interference of the apparent absorbance with other atmospheric absorbers for typical differential CDs for the respective measurement scenario. Table 1 lists the assumed differential CDs of the trace gases absorbing in the same spectral range as the trace gases under investigation. For these potentially interfering trace gases we chose relatively high values, so that the indicated cross interferences correspond to upper limits. The listed CDs represent differential CDs across a typical image FOV ($\sim 20°$), assuming that within the image always a reference region $I_0$ without the target trace gas is present.

In a second step, we calculate the corresponding photon budgets in order to infer the approximate achievable spatial and temporal resolution of the respective imaging measurement.

### 3.1  Description of the Model

The apparent absorbance is calculated from radiances $I_k$ (in $[\mathrm{photons\,s^{-1}\,mm^{-2}\,sr^{-1}}]$) of scattered solar radiation transmitted by the respective spectral channel $k = A, B$ (on-band and off-band FPI setting) :

$$I_k = \int \mathrm{d}\lambda\, I_0(\lambda) e^{-\sum_i \sigma_i(\lambda) S_i} T_{FPI,k}(\lambda) T_{BPF}(\lambda) \tag{7}$$

A spectrum recorded at a clear day in Heidelberg with a solar zenith angle of 73° (160° relative solar azimuth, 89° viewing zenith angle) was used to approximate the spectral radiance $I_0(\lambda)$. $I_0(\lambda)$ was scaled with scattered sky radiance measurements from (Blumthaler et al., 1996). The radiance measurements were performed in Innsbruck in February, 1995 with a solar zenith angle of 68°. For our calculations we used the values for 180° relative solar azimuth and 70° viewing zenith angle. $S_i$ is the

CD of an absorbing gas species $i$ with spectral absorption cross section $\sigma_i(\lambda)$. $T_{FPI,k}$ is the FPI transmittance in configuration $k$ (see Eq. (4)) and $T_{BPF}$ the transmittance of the BPF isolating the measurement wavelength range for the respective target trace gas. $T_{BPF}$ was modeled with a higher order Gaussian function:

$$T_{BPF}(\lambda) = P e^{-\left(\frac{(\lambda_{BPF}-\lambda)^2}{2c^2}\right)^p} \tag{8}$$

with a FWHM of

$$\delta_{\lambda,BPF} = 2c\sqrt{2(\log 2)^{\frac{1}{p}}} \tag{9}$$

$P$ describes the peak transmission at a central wavelength $\lambda_{BPF}$. An order $p = 6$ was used to approximate interference filter transmission profiles.

With Eq. (5) and the intensities $I_k$ from Eq. (7), the apparent absorbance $\tilde{\tau}_i$ can be calculated, allowing to study the sensitivity and selectivity of the detection of a trace gas $i$ for given FPI instrument settings.

In addition, we approximate the respective detection limits based on photon shot noise. In order to calculate the number of photons that reach the detector of the imaging device, we need to know the etendue (product of entrance area $A$ and aperture solid angle $\Omega$) of the employed optics. Kuhn et al. (2014) suggest several optical setups for FPI Correlation Spectroscopy imaging implementations. Here, we chose the setup in which, with help of an image space telecentric optics (see Fig. 5), the incident radiation from the imaging FOV is parallelised before traversing the FPI and BPF. In order to avoid strong blurring of the FPI transmission spectrum due to different incidence angles, the divergence $\Theta$ of the light beams traversing the FPI should not be much larger than $\Theta = 1°$. With a lens of focal length $f$ this condition limits the maximum aperture radius $a$ to:

$$a = f \tan\frac{\Theta}{2} \tag{10}$$

The FPI clear aperture radius $a_{FPI}$ determines the imaging FOV aperture angle

$$\gamma_{FOV} = 2\arctan\frac{a_{FPI}}{f} \tag{11}$$

The etendue per pixel $E_{pix}$ is determined by the spatial resolution of the recorded image, which can be varied by binning individual pixels. For $n_{pix}$ being the number of pixels along a column of a square detector array, the approximate etendue per pixel of the instrument is:

$$E_{pix} = A_{pix}\,\Omega_{pix} \approx a^2 \sin^2(\frac{\gamma_{FOV}}{2n_{pix}})\pi^2 \tag{12}$$

The detectors quantum efficiency and losses within the optics are considered to be not wavelength dependent in the regarded spectral ranges and combined in a loss factor $\eta$. We chose a somewhat lower loss factor for the UV (0.25 for $SO_2$ and $BrO$) compared to the Vis (0.5 for $NO_2$) due to the higher quantum efficiency of commonly used detectors. Each FPI channel (on-band and off-band setting) requires one image acquisition. We assume a photon electron shot noise limited measurement, where for an exposure time $\Delta t$ the number of counted photo electrons per pixel and image is

$$N_{phe,pix} = I\,E_{pix}\eta\Delta t \tag{13}$$

with an uncertainty of $\Delta N_{phe,pix} = \sqrt{N_{phe,pix}}$. The uncertainty in the apparent absorbance $\tilde{\tau}$ is then

$$\Delta \tilde{\tau} \approx \sqrt{\frac{2}{N_{phe,pix}}} \tag{14}$$

assuming the intensities $I_k$ for the two FPI settings $k = A, B$ are similar and that the reference intensities $I_{0,k}$ have to be recorded only once.

Note, that the used sky radiances, loss factors and dimensions of the optics (see Tab. 2) represent conservative assumptions. For instance the light throughput could be enhanced by more than an order of magnitude by choosing a different optical setup (see Kuhn et al., 2014). There, the FPI is placed in front of the lens using the full clear aperture and the full aperture angle of the FPI and the optics. Each viewing direction of the FOV, however, will have a different incidence angle onto the FPI and therefore a different FPI transmission spectrum, which has to be accounted for in the data analysis. Alternatively, simply a larger FPI

could be used. The results of the following calculations for the image space telecentric optics, therefore, represent lower limits of the performance.

## 3.2    Results of the simulations

The FPI Correlation Spectroscopy technique allows for numerous different realisations regarding the used spectral window and FPI instrument parameters that can be chosen according to e.g. measurement conditions or availability of optical components

(FPI, BPF). Here, we identified spectral windows in which the target trace gas absorption cross sections exhibit approximately periodic structures and appropriate FPI parameters were determined in order to maximise the correlation of FPI transmission and trace gas absorption according to the procedures described in Kuhn et al. (2014). Table 2 lists the parameters for the exemplary setups we use in this work.

The results for $SO_2$, BrO and $NO_2$ are summarized in Fig. 2, 3 and 4, which show the differential optical densities of the

target trace gas and the potentially interfering trace gases for the respective measurement wavelength ranges (panels a). In the lower panels, the transmitted spectral radiances of the respective FPI spectral channels (on-band, off-band) are plotted. The $SO_2$ and $NO_2$ trace gas optical density clearly dominate the total differential optical density for the targeted detection limits. For BrO the other trace gases exhibit differential optical depths on the same order of magnitude as BrO. In the b) panels of the same figures, the respective simulated calibration curves are plotted, where the dashed lines indicate the impact of the

individual interferring gases for the assumed amounts. For all three gases these impacts are well below the targeted detection limit. Especially for the case of BrO this illustrates how FPI Correlation Spectroscopy can effectively separate the absorption structure of a single trace gas from a multitude of trace gas optical densities of the same order of magnitude. By using more than two FPI settings, the selectivity can be enhanced even further.

The absorption bands of $NO_2$ are not ideally periodic in the chosen wavelength window. Therefore, at first glance they appear

to be non-ideal for FPI correlation. The apparent absorbance, however, is still reasonably high with extremely low cross interferences to water vapour and $O_4$. This demonstrates, that periodical absorption structures are ideal but not necessary for FPI Correlation Spectroscopy. For a different measurement scenario (here we optimised for stack and ship emissons, see above) there might also be a better choice for instrument parameters. For instance for a high radiance and low $NO_2$ scenario one might

use a single FPI transmission peak on the $NO_2$ absorption band at $\sim 435\,nm$ (as setting A) and at $\sim 438\,nm$ (as setting B) in order to increase the sensitivity (see Fig. 4).

Table 3 summarises the results of the photon budget calculations. We calculated the maximum possible spatial resolution of the imaging measurement for a $10\,s$ exposure time and the instrument parameters listed in Tab. 2. For this, spatial pixels are co-added until the targeted detection limit was reached. We find that for the targeted detection limits the spatial resolutions of the imaging measurements for the chosen parameters are 226 by 226 pixels for $SO_2$, 51 by 51 pixels for BrO and 252 by 252 pixels for $NO_2$ for a temporal resolution of 10 s. The temporal resolution could be enhanced at the expense of the spatial resolution or vice versa. For instance, cutting the linear spatial resolution in half (e.g. from 226 by 226 to 113 by 113 pixels for $SO_2$), would reduce the temporal resolution to 5s for the same detection limit.

When comparing to corresponding DOAS measurements the increase of spatio-temporal resolution becomes evident. A state of the art DOAS measurement takes around $1\,s$ to reach a detection limit of $1 \cdot 10^{14}\,molec\,cm^{-2}$ BrO for one spatial pixel. To scan the ca. 2600 pixels of the assumed BrO image would take 2600s. This is, however, a comparison with an instrument that is not optimised for this kind of imaging measurements. Manago et al. (2018) recently recorded $NO_2$ images with a hyperspectral camera, based on imaging DOAS. A detection limit around $1 \cdot 10^{16}\,molec\,cm^{-2}$ $NO_2$ is reached with a spatial resolution of 480 by 640 pixels and 3 by 3 pixel binning with $12\,s\,frame^{-1}$. This is only a factor of 2.2 slower than our calculation for the telecentric setup. By applying the standard optics introduced in Kuhn et al. (2014), the light throughput is increased by another factor of 32. Therefore, theoretically, the FPI technique can be by a factor of $\sim 70$ times faster. Of course these values always depend on the size of the assumed instrument optics. Our results show that FPI Correlation Spectroscopy can be about two orders of magnitude faster than conventional DOAS measurements while maintaining a similar degree of selectivity and interference suppression.

The presented results of the exemplary calculations for $SO_2$, BrO and $NO_2$ suggest that FPI Correlation Spectroscopy can also be implemented for other trace gases with similarly strong and structured absorption, such as e.g. $O_3$, HCHO, IO, or OClO.

## 4 Proof of concept: Field measurements of volcanic $SO_2$

### 4.1 Sensitivity and ozone interference

The above model study on trace gas detection with FPI Correlation Spectroscopy was validated in a proof of concept field study for volcanic $SO_2$. In a one-pixel prototype a single photodiode was used as detector. A BPF ($\lambda_{BPF} \approx 310\,nm, \delta_{\lambda,BPF} \approx 10\,nm$) was used for the preselection of a wavelength range, where the $SO_2$ differential absorption is strong and approximately periodic (see Fig. 2). A FPI (air-spaced etalon from *SLS Optics Ltd.*) with a FSR of 2.1 nm and a Finesse of 7 across a clear aperture of 20 mm was tilted by a servo motor in order to tune it to the on-band and off-band transmission settings. The individual plates of the FPI have a finite thickness and two surfaces, the outer surfaces have an anti-reflective coating and are slightly wedged from the inner surfaces of the plates, so their influence can be neglected here. The optical setup behind FPI and BPF consists of a fused silica lens ($f \approx 50\,mm$), which projects light from a narrow FOV ($\sim 0.8°$ aperture angle) onto the photodiode.

Radiances for the on-band and off-band channel were recorded, delivering an apparent absorbance measurement with 0.42 Hz. A telescope ($\sim 0.5°$ aperture angle) was co-aligned with the one pixel FPI setup and connected to a temperature stabilized spectrometer (spectral resolution $\sim 0.8\,\mathrm{nm}$). The recorded spectra ($\sim 0.13\,\mathrm{Hz}$) were evaluated with the DOAS algorithm.

The measurement was performed at the *Osservatorio Vulcanologico Pizzi Deneri* (37.766° N, 15.017° E, 2800 m a.s.l.) at Mt. Etna on Sicily on 30 July 2017. The device was pointed towards the volcanic plume of Mt. Etna with constant viewing angle (8° viewing elevation, azimuth 280° N). A plume free part of the sky (zenith viewing direction) was used for reference measurements and recorded prior to the plume measurement. Fig. 6 shows the time series of the apparent absorbance of the FPI Correlation Spectroscopy prototype together with the $SO_2$ CD retrieved from the co-recorded spectra. The apparent absorbance shows high correlation with the retrieved $SO_2$ CD. In Fig. 7 the correlation plot is shown. For high $SO_2$ CDs the sensitivity of $\tilde{\tau}_{SO_2}$ decreases slightly due to saturation effects. The scatter of the values mainly originates from slight misalignment and the difference of the two narrow FOVs.

The recorded UV spectra also allow for evaluating the $O_3$ absorption. The lower panel of Fig. 6 shows the change of the differential $O_3$ CD during the measurement with respect to the reference. The observed increase of the $O_3$ CD by more than $4 \cdot 10^{18}\,\mathrm{molec\,cm^{-2}}$ during the plume measurement is due to the increasing stratospheric light path with increasing solar zenith angle (63.58° to 79.31° during the measurement sequence). Within an imaging FOV (of e.g. 17°) much lower differential $O_3$ CDs are expected (see Tab. 2), since all pixels are similarly affected by the change in $O_3$ background. Even with this extreme change in $O_3$ CD no impact on the recorded $SO_2$ apparent absorbances is observed.

The presented data also indicates the potential of using an additional DOAS measurement for the calibration of the apparent absorbance of an FPI imaging device. The position of the narrow FOV of a DOAS telescope pointing into the wide imaging FOV can be retrieved from time series and used for an in-operation calibration (see e.g. Lübcke et al., 2013; Sihler et al., 2017).

## 4.2    Calculation of $SO_2$ CDs by modeling effective absorption cross sections

As stated in Sect 2.2 we can also directly calculate the $SO_2$ CDs from the apparent absorbance $\tilde{\tau}$ by modeling the effective absorption cross sections and thereby $\Delta\overline{\sigma}_{SO_2}$. This requires knowledge about the instrument spectral transmission, the background scattered light spectrum and the $SO_2$ absorption cross section.

We modeled the instrument transfer function with the transmission spectrum of the used band pass filter, the calculated FPI transmission spectrum (see Sect. 3.1) and the quantum efficiency of the photodiode. The background scattered sunlight spectrum was modeled using a high resolution solar atlas spectrum according to Chance and Kurucz (2010), scaled by the wavelength to the fourth power (assuming Rayleigh scattering) and multiplied with the transmission of the total slant atmospheric ozone column. The ozone column was estimated to $2.5 \cdot 10^{19}\,\mathrm{molec\,cm^{-2}}$ using the vertical ozone column (for the measurement day according to satellite measurements, TEMIS database, Veefkind et al., 2006) multiplied with a geometric air mass factor for the average solar zenith angle during the measurement. The $SO_2$ absorption cross section of Vandaele et al. (2009) was used.

The largest uncertainties are the finesse of the FPI and the modeled background spectrum, where Rayleigh scattering approximation and the assumed ozone column introduce uncertainties. A finesse of about 7 is reported by the manufacturer for

perpendicularly incident radiation. For the instrument model we have to calculate the effective finesse for a divergent light ray ($\sim 0.8°$ aperture angle) for the two FPI tilt positions (around $0°$ for setting B and $5°$ for setting A, corresponding to a finesse of around 7 and 5, respectively). Since the divergent light ray reaching the detector is dependent on focal length, detector area, the alignment of the optical components and due to the uncertainty in the reflectance of the FPI we can determine the finesse

only with an uncertainty of $\pm 3\%$. Further we estimate an uncertainty in the background spectrum by $\pm 10\%$ in our calculation, accounting for uncertainties in atmospheric radiative transfer and ozone column.

Figure 8 shows the $SO_2$ CDs calculated with the described FPI model as a function of the $SO_2$ CDs retrieved from the DOAS spectra. We observe an excellent agreement with a slope of 0.99, an intercept of $7.5 \cdot 10^{15}\,\mathrm{molec\,cm^{-2}}$ and $R^2 = 0.96$. The uncertainties in background spectral radiance and finesse result in a uncertainty in the $SO_2$ sensitivity $\Delta\overline{\sigma}_{SO_2}$ of around $\pm 10\%$

and therefore a relative uncertainty of $\pm 10\%$ in the retrieved $SO_2$ CD. Here, it is important to highlight the difference between ozone interference with the apparent absorbance and the influence of an uncertainty in the total ozone column assumed in the model. The former seems to be negligible as shown by the measurements (Fig. 6) and in the model study (Sect. 3.2). The latter influences the modeled sensitivity of the measurement. This means, it introduces a small relative uncertainty to the retrieved CDs, which has almost no influence on the detection limit. The saturation effects observed for high CDs in the apparent ab-

sorbances (see Fig. 7), meaning the CD dependency of $\Delta\overline{\sigma}_{SO_2}$, are accounted for by the model as well. This is a very promising result, pointing towards the possibility of calibration free measurements, which would be another major advantage of the FPI Correlation Spectroscopy compared to e.g. filter based $SO_2$ cameras.

## 4.3   Imaging

The one pixel FPI Correlation Spectroscopy prototype, introduced in this study, can be implemented in a full frame imaging instrument. This is the major advantage of the technique compared with the DOAS technique. The imaging implementation can be achieved with e.g. the image space telecentric optical setup, used for the above calculations and shown in Fig. 5. In principle, the single pixel detector (photodiode) is replaced by a two dimensional detector array (UV sensitive for $SO_2$ and $BrO$) and an aperture stop is added in the focal plane in front of the lens. This would, however, reduce the light throughput

25    per pixel of the imaging setup compared to the one pixel prototype. Alternatively, the FPI could be placed in front of the lens using the full clear aperture and the full aperture angle of the FPI and the optics, increasing the light throughput by a factor of 32 (see Kuhn et al., 2014). This leads to a much higher light throughput, however, the incidence angle of the incident light onto the FPI and thereby the FPI transmission spectrum becomes dependent on the pixel (i.e. the viewing direction within the imaging FOV) and has to be accounted for in the data evaluation.

## 5 Conclusions

Many locally variable atmospheric processes are difficult to quantify with state of the art UV/Vis remote sensing methods (e.g. DOAS) due to the limited spatio-temporal resolution. This makes it difficult to e.g. study the emission of point sources or to separate the effects of transport and chemical conversion on local scales. Kuhn et al. (2014) proposed the FPI Correlation Spectroscopy for $SO_2$ in the UV wavelength range after similar approaches have been studied in infrared wavelength ranges (e.g. Wilson et al., 2007; Vargas-Rodríguez and Rutt, 2009). The major motivation is to reduce the number of spectral channels used for the trace gas detection in order to increase the spatio-temporal resolution of the measurement while maintaining its selectivity.

In a model study we investigated the sensitivity and determined the photon budget of FPI Correlation Spectroscopy for three measurement scenarios for $SO_2$, $BrO$ and $NO_2$. For $SO_2$ we assumed a scenario with rather low volcanic emissions, which is also representative for industrial stack or ship emissions. For $BrO$ a scenario with stronger volcanic emissions was assumed, with $BrO$ mixing ratios of 10 to 100 ppt within the volcanic plume and high $SO_2$ CDs. The $NO_2$ measurement scenario represents typical stack emissions of power plants and gradients of local air pollution induced by e.g. traffic.

For all three investigated gases, cross interferences with other trace gases absorbing in the preselected spectral ranges were found to be very low, meaning that the selectivity of FPI Correlation Spectroscopy can be similar to the selectivity of conventional techniques (e.g. DOAS). In this study, we only used two FPI settings. A larger number of FPI settings could be used to further reduce possible cross interferences.

Using rather conservative assumptions regarding the intensity of the incoming radiation and the size of the instrument optics, we calculated the highest possible spatio-temporal resolution of the FPI Correlation Spectroscopy measurements for the different scenarios and found that they can be more than two orders of magnitude higher compared to state of the art DOAS measurements for the same trace gas CD. This means that in the same time period a conventional dispersive technique records a single viewing direction (i.e. a single spatial pixel), almost an entire image can be recorded with the FPI Correlation Spectroscopy. This strongly indicates that future instruments based on FPI Correlation Spectroscopy can provide unprecedented insight into short time or small scale processes in the atmosphere.

In the second part, we presented a proof of concept field study for FPI Correlation Spectroscopy applied to volcanic $SO_2$, which confirms the model simulations by comparing the measured apparent absorbance to $SO_2$ CDs retrieved by a co-aligned DOAS measurement. One particularly important finding is that, as expected from the model study, no $O_3$ cross interference can be observed over a large $O_3$ CD range. Further, $SO_2$ CDs could directly be calculated from the instrument model and a very simple radiative transfer model very accurately and with a $\sim 10\%$ uncertainty of the sensitivity. This indicates that CDs can be retrieved directly from the FPI radiance data without calibration.

The extension of the one pixel prototype to a camera can be accomplished comparably easily by minor modifications of the optics and by using a UV sensitive detector array and should be the aim of future studies. By replacing the FPI and the BPF, the instrument is adjusted to measure different trace gases, e.g. $BrO$ and $NO_2$ according to the model calculations performed in Sect. 3.

The applications of UV/Vis FPI Correlation Spectroscopy mentioned in this work represent only some examples for trace gases and phenomena that could be studied. Beyond the volcanological application, FPI imaging can for instance be used to study $SO_2$ in air pollution or $BrO$ in salt pans (see e.g. Holla et al., 2015). The technique can also be applied to other trace gases with similarly strong and structured absorption, such as e.g. $O_3$, HCHO, IO, or OClO.

5   *Data availability.*   The data used for the proof of concept study can be obtained from the authors upon request.

*Competing interests.*   The authors declare that they have no conflict of interest.

*Acknowledgements.*   We would like to thank *SLS Optics Ltd.* for sharing their expertise in designing and manufacturing etalons.

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

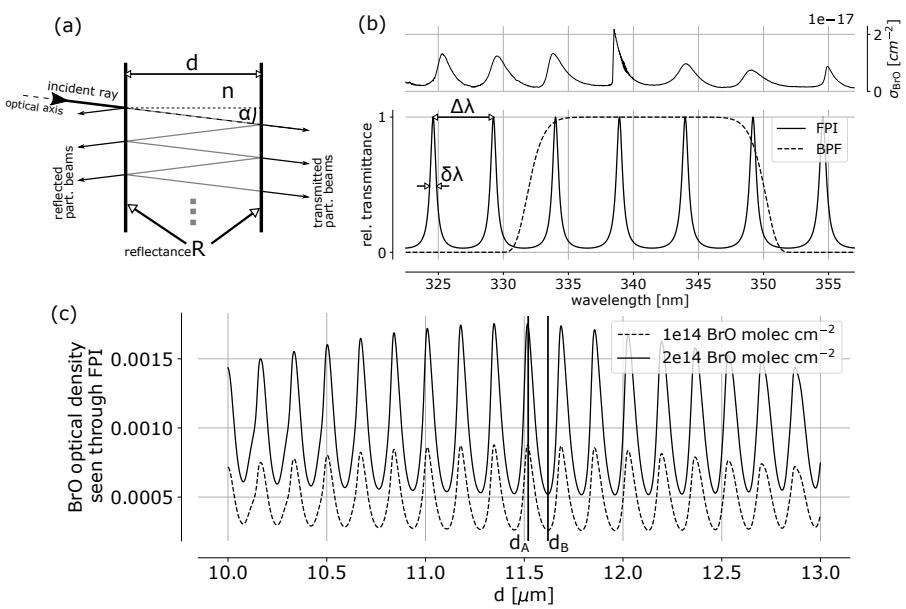

**Figure 1.** (a) FPI schematics indicating the splitting of incident radiation into partial beams that interfere to cause the FPI transmittance spectrum (b), which is characterised by periodic transmission maxima with a FWHM of $\delta_\lambda$ and a FSR of $\Delta_\lambda$. The $\mathrm{BrO}$ absorption cross section (upper panel) shows approximately periodic structures allowing for a high correlation with spectral FPI transmittance. This leads to a modulation of the BrO optical density as seen through the FPI with changing surface displacement $d$ (c). The apparent absorbance is the difference of the optical densities of FPI settings A and B, representing maximum and minimum correlation of FPI transmission and absorption cross section $\sigma$.

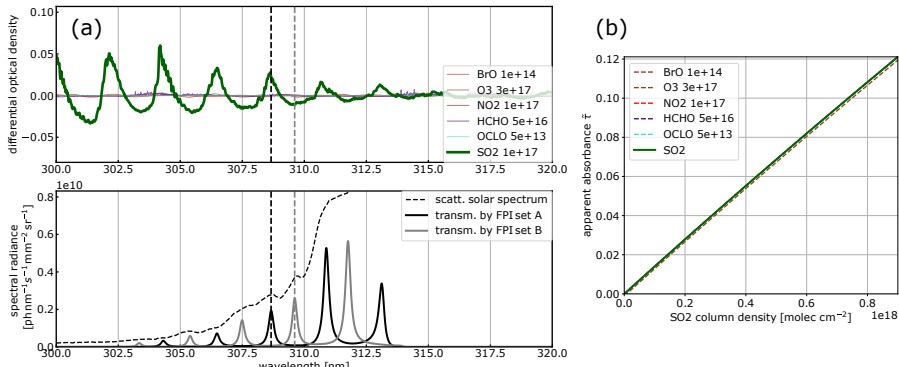

**Figure 2.** Model results for $SO_2$: (a) the differential optical densities of the assumed differential trage gas CDs are plotted in the upper panel. The lower panel shows the spectral radiance of the sky (dashed line) and the transmitted spectral radiances of the FPI and BPF (drawn lines, on-band in black, off-band in gray). (b) shows the calculated calibration curve for $SO_2$ only (drawn line) and with different interfering species included (dashed lines, CDs in $[\mathrm{molec\,cm^{-2}}]$ see legend and Tab. 1).

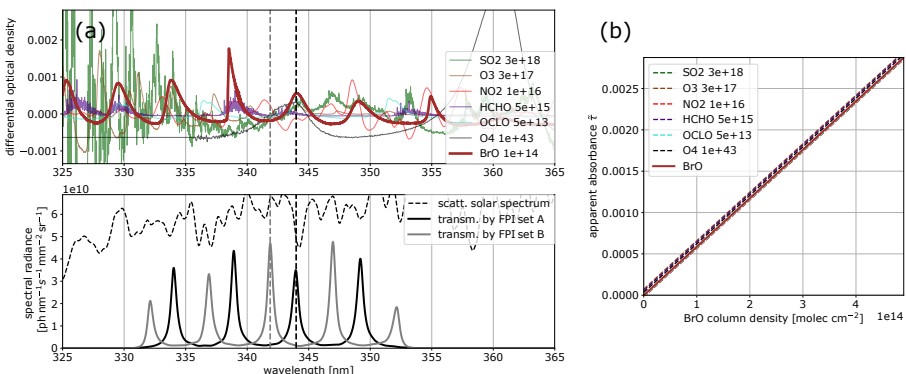

**Figure 3.** Model results for BrO: (a) the differential optical densities of the assumed differential trage gas CDs are plotted in the upper panel. The lower panel shows the spectral radiance of the sky (dashed line) and the transmitted spectral radiances of the FPI and BPF (drawn lines, on-band in black, off-band in gray). (b) shows the calculated calibration curve for BrO only (drawn line) and with different interfering species included (dashed lines, CDs in $[\mathrm{molec\,cm^{-2}}]$ see legend and Tab. 1).

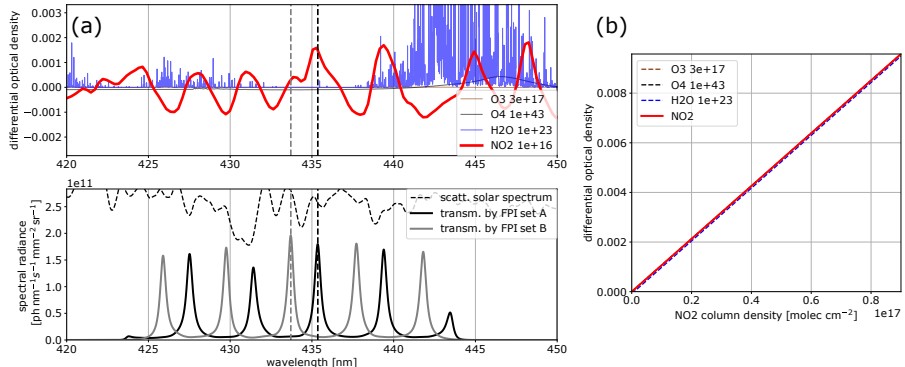

**Figure 4.** Model results for $NO_2$: (a) the differential optical densities of the assumed differential trage gas CDs are plotted in the upper panel. The lower panel shows the spectral radiance of the sky (dashed line) and the transmitted spectral radiances of the FPI and BPF (drawn lines, on-band in black, off-band in gray). (b) shows the calculated calibration curve for $NO_2$ only (drawn line) and with different interfering species included (dashed lines, CDs in $[\mathrm{molec\,cm^{-2}}]$ see legend and Tab. 1).

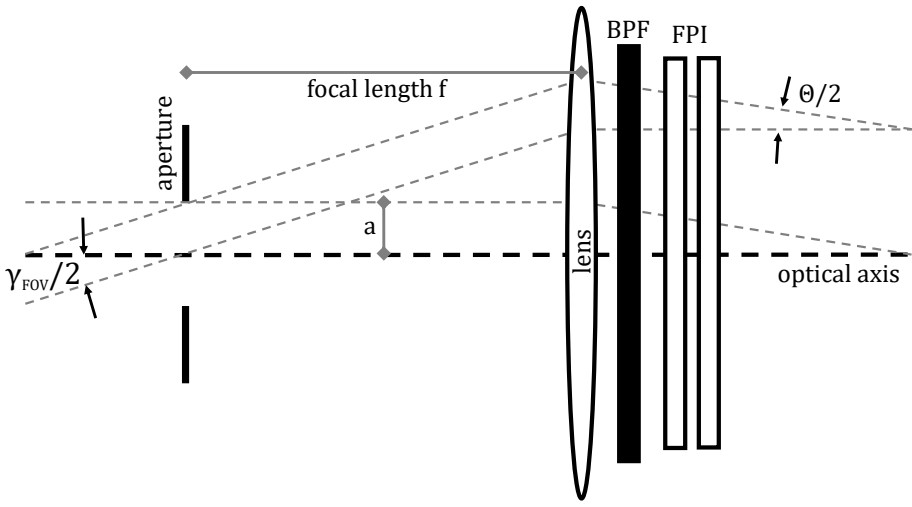

**Figure 5.** Image space telecentric optical setup for parallelising light from the imaging FOV before traversing the FPI and BPF.

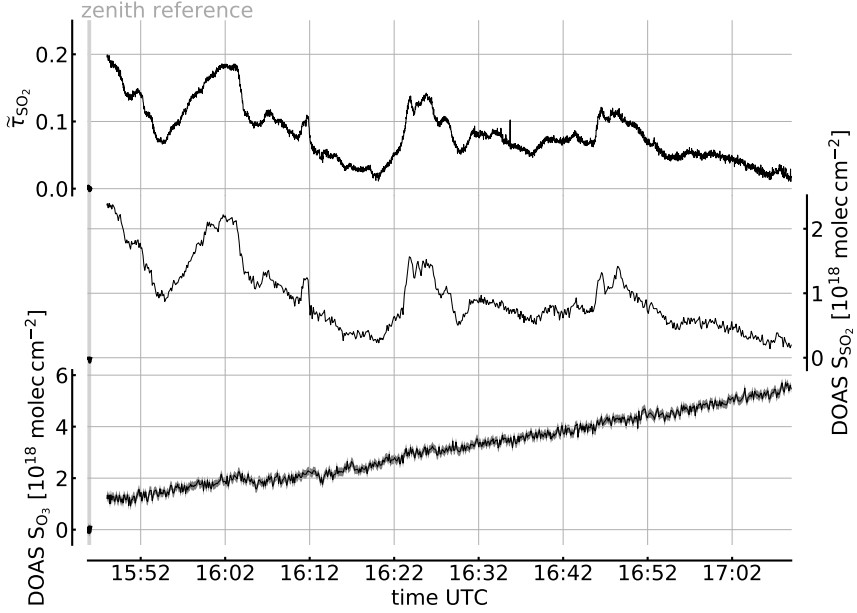

**Figure 6.** Time series of the apparent absorbance of the one pixel FPI Correlation Spectroscopy prototype for $SO_2$ detection (top trace, left scale) recorded at Etna, Sicily on 30 July 2017. A co-aligned telescope was used to simultaneously record spectra for DOAS evaluation of $SO_2$ and $O_3$ (center and bottom traces and right and bottom left scales, respectively). The apparent absorbance nicely correlates with the $SO_2$ CD (see Fig. 7), while no $O_3$ impact is observable. The growth of the retrieved $O_3$ differential CD is expected due to the increasing stratospheric $O_3$ column for increasing solar zenith angle (see text).

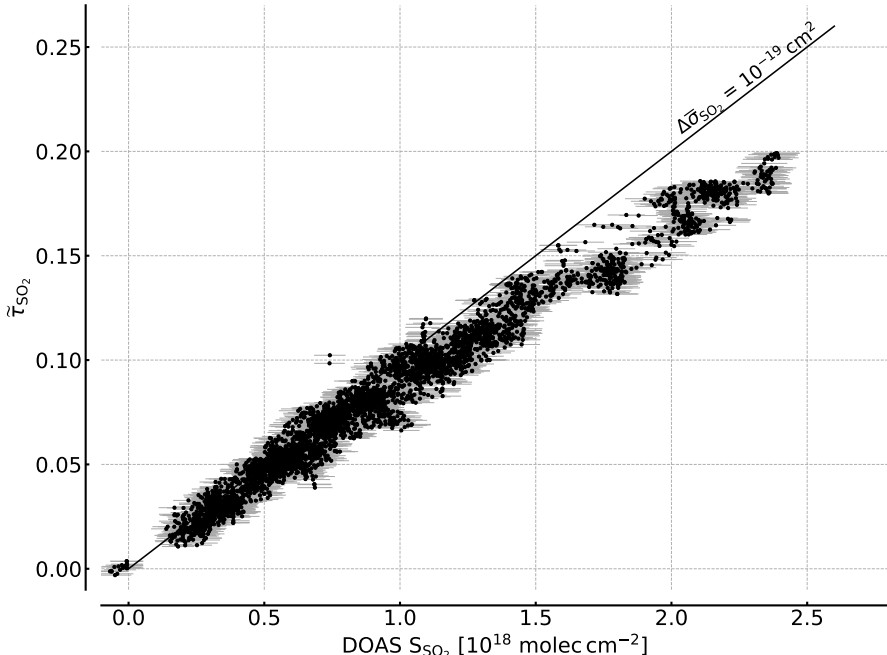

**Figure 7.** Correlation plot of the recorded FPI Correlation Spectroscopy apparent absorbance and the $SO_2$ CD retrieved by DOAS. A sensitivity of about $\Delta\overline{\sigma}_{SO_2}$ of $10^{-19}$ cm$^2$ is reached for lower $SO_2$ CDs. For higher CDs a flattening of the curve is observed that is induced by saturation effects due to the high $SO_2$ optical densities at the absorption peaks.

**Table 1.** Differential CDs assumed for the different measurement scenarios. The values represent high values for CD variations within a typ. imaging FOV. The targeted detection limits are indicated in bold face.

|  | differential CD across imaging FOV [molec cm$^{-2}$] | | |
|---|---|---|---|
|  | $SO_2$: volcanic emission (low) 300 - 315nm | BrO: volcanic emission (high) 330 - 355nm | $NO_2$: stack emission 424 - 445nm |
| $SO_2$ | **1e17** (volcanic) | 3e18 (volcanic) | - |
| BrO | 1e14 (volcanic) | **1e14** (volcanic) | - |
| $NO_2$ | 1e17 (bgr. pollution) | 1e16 (bgr. pollution) | **1e16** (stack plume) |
| $O_3$ | 3e17 (SZA change strat.) | 3e17 (SZA change strat.) | 3e17 (SZA change strat.) |
| HCHO | 5e15 (background) | 5e15 (background) | - |
| $H_2O$ | - | - | 1e23 (background) |
| OClO | 5e13 (volcanic) | 5e13 (volcanic) | - |
| $O_4$ | - | 1e43 ($O_2$ dimer, [molec$^2$cm$^{-5}$]) | 1e43 ($O_2$ dimer, [molec$^2$cm$^{-5}$])) |

typical values based on: Roscoe et al. (2010); Gliß et al. (2015); Bobrowski and Giuffrida (2012); Dekemper et al. (2016); absorption cross sections: Vandaele et al. (2009); Fleischmann et al. (2004); Bogumil et al. (2003); Serdyuchenko et al. (2014); Chance and Orphal (2011); Rothman et al. (2013); Kromminga et al. (2003); Thalman and Volkamer (2013)

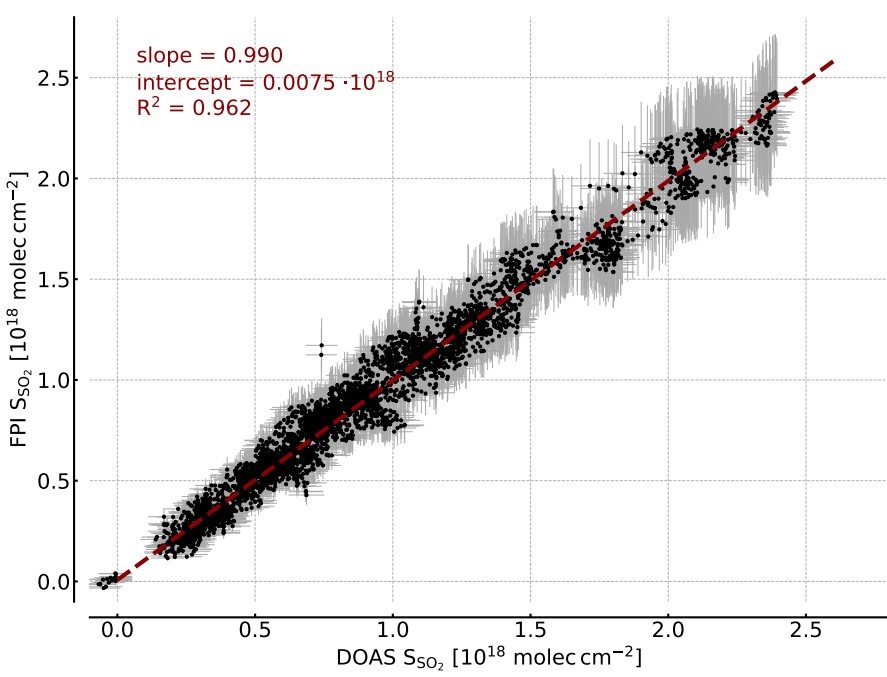

**Figure 8.** Correlation plot of the $SO_2$ CDs, retrieved by modeling and the $SO_2$ CDs retrieved by DOAS. The error bars indicate the uncertainty of the FPI finesse ($\pm 3\%$) and the background spectral radiance in the model ($\pm 10\%$) and the DOAS retrieval error, respectively. A high correlation is observed and the saturation effect is accounted for by the model as well.

**Table 2.** Instrument parameters of FPI, BPF and optical setup used for the simulations. The radiance for the on-band setting $I_A$ at the detector was approximated based on the instrument parameters and sky radiance values from Blumthaler et al. (1996).

| | instrument parameters | | | |
|---|---|---|---|---|
| | SO$_2$ | BrO | NO$_2$ | |
| $d_A$ [μm] | 21.60 | 11.52 | 23.72 | FPI surface displacement setting A |
| $d_B$ [μm] | 21.44 | 11.62 | 23.63 | FPI surface displacement setting B |
| $R$ | 0.7 | 0.7 | 0.7 | FPI surface reflectivity |
| $P_{BPF}$ | 0.7 | 0.7 | 0.7 | BPF peak transmission |
| $\lambda_{BPF}$ [nm] | 308 | 342 | 434 | BPF central wavelength |
| $\delta_{\lambda,BPF}$ [nm] | 10 | 20 | 18 | BPF FWHM |
| $f$ [mm] | | 50 | | focal length of imaging optics |
| $\Theta$ [°] | | 1 | | required parallelisation |
| $a$ [mm] | | 0.44 | | aperture radius of imaging optics |
| $a_{FPI}$ [mm] | | 7.5 | | aperture radius of FPI |
| $\eta$ | 0.25 | | 0.5 | loss factor |
| $\gamma_{FOV}$ [°] | | 17 | | imaging FOV of camera |
| $I_A$ [photons s$^{-1}$ mm$^{-2}$ sr$^{-1}$] | 4.51e9 | 1.48e11 | 5.17e11 | |

**Table 3.** Simulation results: the spatial resolution of a FPI Correlation Spectroscopy measurement was calculated for an exposure time of 10 s and the target detection limits, shown in Tab. 1.

| | simulation results | | |
|---|---|---|---|
| | SO$_2$ | BrO | NO$_2$ |
| $\Delta\overline{\sigma}$ [cm$^{-2}$] | 1.5e-19 | 6e-18 | 1.1e-19 |
| target det. lim. [molec cm$^{-2}$] | 1e17 | 1e14 | 1e16 |
| target det. lim $\tilde{\tau}$ | 0.015 | 0.0006 | 0.0011 |
| required $N_{phe,pix}$ | 8.9e3 | 5.6e6 | 1.7e6 |
| required $E_{pix}$ [mm$^2$ sr] | 7.9e-7 | 1.5e-5 | 6.6e-7 |
| **max. spatial resolution ($n_{pix} \times n_{pix}$)** | **226×226** | **51×51** | **252×252** |