# Peer review of "Towards imaging of atmospheric trace gases using Fabry Pérot Interferometer Correlation Spectroscopy in the UV and visible spectral range"

_Atmospheric Measurement Techniques, 2018_

## Referee Comment (RC1) · E. Dekemper (Referee) · 12 Nov 2018

Dear Editor,

You gave me the opportunity to review the manuscript by Mr Jonas Kuhn et al. about the progresses made in the development of a new type of instrument dedicated to the remote sensing of UV-VIS absorbing species. My general appreciation is that this work constitutes a significant milestone towards the realization of new instruments capable of following the emission and the dispersion of important trace species (such as $SO_2$, BrO, and $NO_2$) in the atmosphere with a high spatial and temporal resolution. Compared to a previous concept paper by the same author (Kuhn et al, Atmos. Meas. Tech.

[Figure]

7, 2014), the new results described here provide some confidence about the sensitivity of the future instrument thanks to realistic simulations, and field experiments with a non-imaging prototype.

I only have a small number of comments and questions which will, would they be addressed by the author, improve in my view the quality of the manuscript further. In addition I listed some typos.

Specific comments/questions:

- It is true that the concept of aligning the transmission comb of the FPI with the periodic structures of the absorption cross-section of species has been described by the author in an earlier paper. However, I would have liked to see a bit more description here as well. For instance, the radiometric model of the measurements described in section 2.2 should make clearer that the measured signals $I_A$ or $I_B$ are made out of photons captured by each transmission peak of the FPI at the same time. Equation (5) could emphasize this by expressing $I$ as the result of this summation process. Alternatively, one could bring equation (6) into section 2.2.

- Figures 2, 3, and 4 show a possible setting of the FPI targetting a different species respectively. One can imagine that the performance of the measurement method is strongly determined by the capability of finding the best "position" of the FPI comb in order to maximize the correlation. I found that this could have been pointed out and discussed in more details, by showing for instance how evolves this correlation as a function of a spectral displacement of the comb. For instance, by looking at figure 4 (NO2), one observes that one of the FPI peaks of the off-channel actually captures a relatively high absorption. This sounds like a sub-optimal configuration whose impact could have been dealt with.

- The determination of the CD eventually depends on the knowledge of the effective differential cross-section as shown in eq. (5). However, the way to determine this quantity is not discussed. With the actual instrument, will there be sufficient knowledge of the FPI transmission curve and of the BPF to compute it from a high resolution cross-section dataset? Or will it be necessary to measure it in the lab?

- The field experiment shows encouraging results, although it does not go up to comparing SCDs retrieved by the FPI-based instrument with the ones obtained from the classical grating-based instrument. As a reader, one immediately wonders why it is like that. Is it because there is no a priori knowledge of the effective differential cross-section (see point before), or because no clear sky measurement could be made at the time of the measurements?

- In the discussion of the results of the field experiment, there is time spent on the interference by O3. However, in absence of any attempt to retrieve the SO2 CD from the FPI-based measurements, it is difficult to adhere to the conclusions of the author about a relative insensitivity to O3. On the other side, if this sensitivity is more robustly confirmed, this aspect is an important asset compared to the widespread SO2 cameras, and emphasized further.

- In section 3, there are some inconsistancies between the text and Table 1 which it is referring to. Furthermore, the paragraph is not making it completely clear that the selected CDs are actually detection limits. In particular, for NO2, a value of 1e16 is clearly not the maximum that can be observed above a smokestack, hence it must be a detection limit... This is however better stated in the caption of Table 1.

Technical corrections:

- p.2,L.21: repeated "by column"

- p.2,L.22: replace "rather high" by "rather large" or "rather long"

- p.2,L.27: add a comma after "imaging tachniques"

- p.3,L.5: replace "Due to" by "Thanks to"

- p.3,L.11: add commas before and after "for the three gases"

- p.3,L.12: replace "by around" by "about"

- p.4,L.1: remove "a stable"

- p.5,L.28: "calculated" is misspelled

- p.7,L.15: remove the comma after "this illustrates"

- p.10,L.6-7: the end of the sentence is very clear.

---

## Referee Comment (RC2) · Anonymous Referee #2 · 20 Dec 2018

This paper mostly presents a theoretical investigation of the applicability of Fabry Perot interferometers (FPIs) for the fast imaging of atmospheric trace gases, of high relevance for the visualization and quantitative assessment of localized emission sources of molecules such as SO2, BrO and NO2. The study is essentially a follow-up from study by Kuhn et al. (2014) already published in AMT and where the principle and advantages of the approach are described. Here, the authors concentrate on simulations aiming at demonstrating the selectivity of the FPI correlation spectroscopy for three potential target gases: SO2, BrO and NO2. In comparison to more traditional atmospheric trace gas imaging techniques which generally only use a set of band pass filters, the FPI correlation spectroscopy is found to be highly selective, allowing to dras-

tically reduce cross interferences with other absorbing species while maintaining high sensitivity. In short, the method seems to be very promising. The approach used for the simulations is convincing and clearly described.

In my opinion, the manuscript is overall of excellent quality, well written and concise. However, the proof of concept presented at the end of the study is a bit disappointing. One would have expected to see first results from real images. Instead only a one-pixel prototype is shown, and only applied to SO2 retrieval (which arguably is the easiest case). It is claimed that the one-pixel prototype can "easily" be transferred in a full-frame imaging instrument, however this is not demonstrated. At minimum, the authors should explain why this transfer was not attempted in the present study and what are the possible difficulties that may need to be addressed.

Despite this weakness, I think that the study remains very interesting, show a good degree of innovation (first demonstration of the potential of the technique for other species than SO2) and therefore it should be published in AMT.

In fact, my only serious reservation, concerns the estimation of the added performance of the FPI correlation spectroscopy in comparison to a traditional hyperspectral imaging system (as used for imaging DOAS). Based on existing CMOS or CCD detectors coupled to grating spectrometers (commonly used in the DOAS community), and assuming hyperpixels of 100 micron size, experience shows that typically 1.5 Me-/sec/pixel can be accumulated in the visible range under normal illumination conditions. On this basis, one can estimate that a full image of 100 x 100 spatial pixels (or even 200 x 200) could potentially be recorded in a few seconds of time (using e.g. an integration time of 50 msec/hyperspectral line) leading to a S/N ratio of approximately 250 (corresponding to a NO2 dSCD uncertainty of about 5e15 molec/cm2). This level of performance is in fact very close to the performance announced in Table 3, and therefore I believe that the factor 100 announced in the text is rather optimistic (at least for a FPI system using a telecentric optics as assumed here). Of course the real proof will have to wait for actual measurements and I look forward to see more of such measurements being

attempted and published in the near future.

---

## Author Comment (AC1) · 17 Jan 2019

Author's response to Review RC1 by Emmanuel Dekemper

We are very grateful for the constructive comments of Emmanuel Dekemper, which led to a very valuable extension of the manuscript. As a major point, we added the retrieval of SO2 CDs from apparent absorbances determined with the prototype instrument, which indicates the possibility of calibration free measurements.

In the following, we will answer the individual comments. For clarity we numbered them from 1) to 6) (bold font). The original reviewer comment is set in italic font, the authors' response in normal font.
We add the two sections (Sect 2.2 and Sect 4) and four Figures (1, 6, 7, 8), containing the major changes to the end of this file.

*Specific comments/questions:*

**1)**

*It is true that the concept of aligning the transmission comb of the FPI with the periodic structures of the absorption cross-section of species has been described by the author in an earlier paper. However, I would have liked to see a bit more description here as well. For instance, the radiometric model of the measurements described in section 2.2 should make clearer that the measured signals $I_A$ or $I_B$ are made out of photons captured by each transmission peak of the FPI at the same time. Equation (5) could emphasize this by expressing $I$ as the result of this summation process. Alternatively, one could bring equation (6) into section 2.2.*

This is a good point. We extended Section 2.2 by a new Equation (Eq. 6), clarifying that $I_A$ and $I_B$ are spectral radiances integrated over the FPI spectral transmission interval. Furthermore, we added Fig. 1c in order to clarify the detection principle and also to visualize the optimisation of the instrument parameters for the individual trace gases (also regarding comment 2a).

**2)**

**a)** *Figures 2, 3, and 4 show a possible setting of the FPI targeting a different species respectively. One can imagine that the performance of the measurement method is strongly determined by the capability of finding the best "position" of the FPI comb in order to maximize the correlation. I found that this could have been pointed out and discussed in more details, by showing for instance how evolves this correlation as a function of a spectral displacement of the comb.*

**b)** *For instance, by looking at figure 4 (NO2), one observes that one of the FPI peaks of the off channel actually captures a relatively high absorption. This sounds like a suboptimal configuration whose impact could have been dealt with.*

a) This point is largely covered by our response to comment 1). The best "position" ($d_A$ and $d_B$) are also indicated in Fig. 1c, representing the strongest optical density modulation.

b) Indeed the correlation of optimized FPI transmission spectrum and spectral trace gas absorption doesn't look optimal to the eye for the case of $NO_2$. This is because, the strong absorption band of $NO_2$ at around 425 to 450 nm are not equidistant. This results in a lower sensitivity, compared for example to a measurement setup where a single FPI transmission

peak is isolated and tuned to maximum and minimum absorption (e.g. at 435nm and 438nm). However, regarding the signal to noise ratio, which is also dependent on the total transmission of the instrument and cross interferences with other gases our proposed setup shows a optimal performance for our boundary conditions. There might be different optimal settings for other scenarios (e.g. a scenario with low $NO_2$ levels but high scattered sunlight radiances).    We added the following explanation to Sect. 3.2:

'The absorption bands of $NO_2$ are not ideally periodic in the chosen wavelength window. Therefore at first glance they appear to be non-ideal for FPI correlation. The apparent absorbance, however, is still reasonably high with extremely low cross interferences to water vapour and $O_4$. This demonstrates, that periodical absorption structures are ideal but not necessary for FPI Correlation Spectroscopy. For a different measurement scenario (here we optimised for stack and ship emissions, see above) there might also be a better choice for instrument parameters. For instance for a high radiance and low $NO_2$ scenario one might use a single FPI transmission peak on the $NO_2$ absorption band at ~435 nm (as setting A) and at ~438nm (as setting B) in order to increase the sensitivity (see Fig. 4).'

**3)**

*The determination of the CD eventually depends on the knowledge of the effective differential cross-section as shown in eq. (5). However, the way to determine this quantity is not discussed. With the actual instrument, will there be sufficient knowledge of the FPI transmission curve and of the BPF to compute it from a high resolution cross-section dataset? Or will it be necessary to measure it in the lab?*

Depending on the knowledge of instrument parameters the instrument function can be modelled and used, together with a simple atmospheric radiation model, to determine CDs from apparent absorbances. We added an explanation to Sect. 2.2 and performed the CD calculation for our prototype data in Sect. 4.2. (see response to comment 4).

**4)**

*The field experiment shows encouraging results, although it does not go up to comparing SCDs retrieved by the FPI-based instrument with the ones obtained from the classical grating-based instrument. As a reader, one immediately wonders why it is like that. Is it because there is no a priori knowledge of the effective differential cross-section (see point before), or because no clear sky measurement could be made at the time of the measurements?*

This is a very good comment that led us to calculate $SO_2$ CDs from the apparent absorbance data, without the use of data provided by the co-aligned DOAS spectrograph. The possibility of calibration free measurements that could be shown to work pretty well for our $SO_2$ study constitutes another substantial advantage of our FPI-technique.
Section 4 was subdivided into 3 parts: In '4.1 Sensitivity and ozone interference' the proof of concept study is introduced and the correlation of apparent absorbance with DOAS CDs as well as the ozone interference are presented. In '4.2 Calculation of $SO_2$ CDs by modelling effective absorption cross sections' the model is described and the results of the retrieved FPI CDs are validated with the DOAS data. The outlook to the imaging application is given in '4.3 Imaging'.
In addition to the changes in Section 4, we added the following sentence at the end of the introduction:

'Further, we show that $SO_2$ CDs can be accurately retrieved from the recorded data without calibration (Sect. 4).'

And we added the explanatory text to the conclusions:

'Further, $SO_2$ CDs could directly be calculated from the instrument model and a very simple radiative transfer model very accurately and with a ~10% uncertainty of the sensitivity. This indicates that CDs can be retrieved directly from the FPI radiance data without calibration.'

In order to further clarify the measurement, we included the reference (zenith sky) measurement into the time series in Fig. 6. It was taken a few minutes before the plume measurement.
Finally, we removed the dashed grey line labelled with 'simulation' in Fig. 7 and added the result of the FPI CD calculations in an additional Fig. 8.

**5)**

*In the discussion of the results of the field experiment, there is time spent on the interference by O3. However, in absence of any attempt to retrieve the SO2 CD from the FPI-based measurements, it is difficult to adhere to the conclusions of the author about a relative insensitivity to O3. On the other side, if this sensitivity is more robustly confirmed, this aspect is an important asset compared to the widespread SO2 cameras, and emphasized further.*

We address this point in our response to comment 4) and by the related changes in the manuscript.

**6)**

*In section 3, there are some inconsistencies between the text and Table 1 which it is referring to. Furthermore, the paragraph is not making it completely clear that the selected CDs are actually detection limits. In particular, for NO2, a value of 1e16 is clearly not the maximum that can be observed above a smokestack, hence it must be a detection limit... This is however better stated in the caption of Table 1.*

This is true. We added 'required detection limit of' in the brackets containing the detection limits in the text in Sect. 3. We also corrected the $NO_2$ CD assumed for stack emission $NO_2$ cross sensitivities of 1e16 to 1e17 molec cm$^{-2}$ in the text. The simulation was done for 1e17 molec$^{-2}$ (see Tab. 1 and Fig. 2).
We additionally changed the $NO_2$ CD in the visualisation of the differential optical density in Fig. 4a from 1e17 to 1e16 molec cm$^{-2}$. That way, all the Figures (Fig. 2, 3, 4) visualise the spectral differential optical density of the target trace gas for the respective target detection limit.

*Technical corrections:*
*• p.2,L.21: repeated "by column"*     corrected as proposed
*• p.2,L.22: replace "rather high" by "rather large" or "rather long"*  corrected as proposed
*• p.2,L.27: add a comma after "imaging techniques"*        corrected as proposed
*• p.3,L.5: replace "Due to" by "Thanks to"*   corrected as proposed
*• p.3,L.11: add commas before and after "for the three gases"*      corrected as proposed
*• p.3,L.12: replace "by around" by "about"*   corrected as proposed

- *p.4,L.1: remove "a stable"*   corrected as proposed
- *p.5,L.28: "calculated" is misspelled*          corrected as proposed
- *p.7,L.15: remove the comma after "this illustrates"*          corrected
- *p.10,L.6-7: the end of the sentence is very clear.*  corrected: removed 'and' before 'trace gases'

We performed the suggested technical corrections.

The following Sections and Figures were revised substantially, mainly due to comment 1, 3) and 4):

[revised manuscript text omitted]
 even the saturation effects are accounted for by the model, which indicates the potential of calibration free measurements.

---

## Author Comment (AC2) · 17 Jan 2019

Author's response to Review RC2 by Anonymous Referee #2

We kindly thank the anonymous referee for the helpful comments. In the following, we repeat the reviewer comment in italic font and answer in regular font at points we find appropriate.

*This paper mostly presents a theoretical investigation of the applicability of Fabry Perot interferometers (FPIs) for the fast imaging of atmospheric trace gases, of high relevance for the visualization and quantitative assessment of localized emission sources of molecules such as SO2, BrO and NO2. The study is essentially a follow-up from study by Kuhn et al. (2014) already published in AMT and where the principle and advantages of the approach are described. Here, the authors concentrate on simulations aiming at demonstrating the selectivity of the FPI correlation spectroscopy for three potential target gases: SO2, BrO and NO2. In comparison to more traditional atmospheric trace gas imaging techniques which generally only use a set of band pass filters, the FPI correlation spectroscopy is found to be highly selective, allowing to drastically reduce cross interferences with other absorbing species while maintaining high sensitivity. In short, the method seems to be very promising. The approach used for the simulations is convincing and clearly described.*

*In my opinion, the manuscript is overall of excellent quality, well written and concise. However, the proof of concept presented at the end of the study is a bit disappointing. One would have expected to see first results from real images. Instead only a one-pixel prototype is shown, and only applied to SO2 retrieval (which arguably is the easiest case). It is claimed that the one-pixel prototype can "easily" be transferred in a fullframe imaging instrument, however this is not demonstrated. At minimum, the authors should explain why this transfer was not attempted in the present study and what are the possible difficulties that may need to be addressed.*

We were focussing our study on the applicability of the FPI-technique to further trace gases, rather than to the implementation of an imaging prototype (although the latter goal still has high priority in our research). The proof of concept was carried out with a one pixel prototype because it can easily be coaligned with a DOAS instrument for validation and thereby allows straightforward and robust validation of the presented technique.

*Despite this weakness, I think that the study remains very interesting, show a good degree of innovation (first demonstration of the potential of the technique for other species than SO2) and therefore it should be published in AMT.*

*In fact, my only serious reservation, concerns the estimation of the added performance of the FPI correlation spectroscopy in comparison to a traditional hyperspectral imaging system (as used for imaging DOAS). Based on existing CMOS or CCD detectors coupled to grating spectrometers (commonly used in the DOAS community), and assuming hyperpixels of 100 micron size, experience shows that typically 1.5 Me-/sec/pixel can be accumulated in the visible range under normal illumination conditions. On this basis, one can estimate that a full image of 100 x 100 spatial pixels (or even 200 x 200) could potentially be recorded in a few seconds of time (using e.g. an integration time of 50 msec/hyperspectral line) leading to a S/N ratio of approximately 250 (corresponding to a NO2 dSCD uncertainty of about 5e15 molec/cm2). This level of performance is in fact very close to the performance announced in Table 3, and therefore I believe that the factor 100 announced in the text is rather optimistic (at least for a FPI system using a telecentric optics as assumed here). Of course the real proof will have to wait*

*for actual measurements and I look forward to see more of such measurements being attempted and published in the near future.*

We thank the reviewer for this valuable comment. The 'two orders of magnitude' are mentioned in a rough comparison of the BrO simulation results for the FPI technique and an approximate exposure time of a common MAX-DOAS setup (which is not optimised for this kind of imaging measurement). Manago et al. (2018) recently published an article introducing hyperspectral imaging measurements of $NO_2$. They achieve a detection limit of ~1e16molec $cm^{-2}$ in 12s exposure with a spatial resolution of 480x640 pixels, binned 3 by 3 (i.e. effectively 160x213 pixels). This is indeed only a factor 2.2 slower than we calculated for our telecentric setup for the FPI technique. If we applied the non-telecentric setup introduced in Kuhn et al. (2014) we would increase the light throughput by a factor of 32 and therefore end up with a ~70 times faster setup.

To make this point clearer in the manuscript we made the following changes:

In the introduction, we added the reference of Manago et al. (2018) and the sentence:

'Recently, Manago et al. (2018) reported $NO_2$ measurements with an hyperspectral camera based on the Imaging DOAS technique with considerably higher spatial resolution (~0.08 Hz).'

We changed the last part in Sect. 3.2 to:

'When comparing to corresponding DOAS measurements the enormous increase of spatio-temporal resolution becomes evident. A state of the art DOAS measurement takes around 1s to reach a detection limit of $1x10^{14}$ molec $cm^{-2}$ BrO for one spatial pixel. To scan the ca. 2600 pixels of the assumed BrO image would take 2600s. This is, however, a comparison with an instrument that is not optimised for this kind of imaging measurements. Manago et al. (2018) recently recorded $NO_2$ images with a hyperspectral camera, based on imaging DOAS. A detection limit around $1x10^{16}$ molec $cm^{-2}$ $NO_2$ is reached with a spatial resolution of 480 by 640 pixels and 3 by 3 pixel binning with 12 s $frame^{-1}$. This is only a factor of 2.2 slower than our calculation for the telecentric setup. By applying the standard optics introduced in Kuhn et al. 2014, (i.e. not a telecentric optics), the light throughput is increased by another factor of 32. Therefore, theoretically, the FPI technique can be by a factor of ~70 times faster. Of course this values always depends on the size of the assumed instrument optics. Our results show that FPI Correlation Spectroscopy can be about two orders of magnitude faster than conventional DOAS measurement while maintaining a similar degree of selectivity and interference suppression.
The presented results of the exemplary calculations for $SO_2$, BrO and $NO_2$ suggest that FPI Correlation Spectroscopy can also be implemented for other trace gases with similarly strong and structured absorption, such as e.g. $O_3$, HCHO, IO, or OClO.'

We also mentioned the above factor of 32 in Sect 4 of our manuscript:

'Alternatively, the FPI could be placed in front of the lens using the full clear aperture and the full aperture angle of the FPI and the optics, increasing the light throughput by a factor of 32 (Kuhn et al., 2014).'